# Density Functional Theory Study on NiN_x_ (x = 1, 2, 3, 4) Catalytic Hydrogenation of Acetylene

**DOI:** 10.3390/molecules27175437

**Published:** 2022-08-25

**Authors:** Cuili Hou, Lihua Kang, Mingyuan Zhu

**Affiliations:** 1College of Chemistry & Chemical Engineering, Yantai University, Yantai 264005, China; 2College of Chemistry & Chemical Engineering, Shihezi University, Shihezi 832000, China

**Keywords:** acetylene hydrogenation, density functional theory, graphene catalyst, DFT-D3

## Abstract

In this study, using the application of density functional theory, the mechanism of graphene-NiN_x_ (x = 1, 2, 3, 4) series non-noble metal catalysts in acetylene hydrogenation was examined under the B3LYP/6-31G** approach. With the DFT-D3 density functional dispersion correction, the effective core pseudopotential basis set of LANL2DZ was applied to metallic Ni atoms. The reaction energy barriers of NiN_x_ catalysts are different from the co-adsorption structure during the catalytic hydrogenation of graphene-NiN_x_ (x = 1, 2, 3, 4). The calculated results showed that the energy barrier and selectivity of graphene-NiN_4_ for ethylene production were 25.24 kcal/mol and 26.35 kcal/mol, respectively. The low energy barrier and high activity characteristics showed excellent catalytic performance of the catalyst. Therefore, graphene-NiN_4_ provides an idea for the direction of catalytic hydrogenation.

## 1. Introduction

In modern industry, ethylene in petrochemical products is one of the mainstream industrial raw materials and plays a critical role in the national economy. Ethylene is also a crucial industrial source and fundamental synthetic monomer [1]. After steam cracking of petroleum hydrocarbons, a large amount of ethylene and a small fraction of acetylene are generated [2]. Since the existence of these acetylenes seriously affects the subsequent polymerization process of ethylene and destroys the polymerization of ethylene, the phenomenon of catalyst deactivation will occur, and the catalytic performance will be reduced [3,4]. It can be seen that selective hydrogenation and elimination of acetylene technology plays an important role in the ethylene industry. Since the reaction process of ethylene hydrogenation to ethane also exists, it is necessary to develop a high catalytic performance catalyst for acetylene hydrogenation.

There is a lot of research in the field of catalytic acetylene hydrogenation on noble metal and non-noble metal catalysts at present. Ball et al. [5] modified AgPd and CuPd bimetallic catalysts with TiO_2_ and employed STEM (scanning transmission electron microscopy), FTIR (Fourier-Transform Infrared, FTIR), and other means to examine the catalysts’ catalytic performance. The results reveal that the catalyst is capable of the selective hydrogenation of acetylene in an ethylene-rich environment under mild conditions. Among them, 100% ethylene selectivity was observed on CuPd_0.08_/TiO_2_ catalyst. Choudhary et al. [6] performed a detailed investigation on the hydrogenation conversion of acetylene over Au/TiO_2_, Pd/TiO_2,_ and Au-Pd/TiO_2_ catalyst. The results revealed that the Au/TiO_2_ catalyst (average particle size: 4.6 nm) synthesized using reduction-oxidation was discovered to exhibit high selectivity for the synthesis of ethylene through temperature programming, but the acetylene conversion activity and catalyst stability were not satisfactory. Noble metal catalysts are expensive and rare. Thus, researchers turned their attention to the field of non-precious metal catalysts, such as Ni, Cu, Ni, Zn, etc. Zhuo et al. [7] examined the use of density in defective graphene (DG)-supported single-atom catalysts (SAC), M_1_/SV-G, and M_1_/DV-G (M = Ni, Pd, and Pt) using density functional theory (DFT). The results reveal that among them, Pt_1_/DV-G and Ni_1_/DV-G are the suitable catalysts for acetylene hydrogenation, and Ni_1_/DV-G shows higher selectivity for the semi-hydrogenation products of acetylene. Riley et al. [8] obtained optimal performance for the selective hydrogenation of acetylene by evaluating different synthetic approaches. It was discovered that doping nickel into the ceria lattice resulted in 100% acetylene conversion while hindering ethane formation. Abdollahi et al. [9] demonstrated that copper nanoclusters can be employed as catalysts for the hydrogenation of acetylene and ethylene using DFT calculations. Additionally, graphene-supported Cu_11_ nanoclusters were selected as catalysts, and it was discovered that they have good activity and selectivity for acetylene hydrogenation. Researchers decided to replace different supports to enhance the reactivity and selectivity of non-noble metal catalysts.

Graphene research has shown a linear growth tendency since it reached people’s field of vision [10]. Graphene is widely employed in flexible electronics, battery materials, catalysts, humidity sensors, and other fields [11,12,13,14,15,16]. If graphene is employed directly as a catalyst, its activity is poor. In particular, the use of heteroatoms or metal atoms doping on graphene has been investigated, and electrocatalysts with M-N_x_ as active centers have recently been reported [17,18,19,20,21,22,23,24,25]. We discovered that the graphene catalysts modified using the above approaches promoted the reactions. Recently, numerous studies have been performed on the catalytic activity of NiN_x_ (x = 1, 2, 3, 4) in various reactions [26,27,28,29,30,31,32,33,34,35], but no research on the catalytic hydrogenation of acetylene has been reported. Inspired by the above research work and experimental findings, we introduced N atoms as doping atoms in graphene to construct NiN_x_ (x = 1, 2, 3, 4) series catalysts. The aim of this paper is to investigate the reaction pathway of catalytic hydrogenation of acetylene based on the reaction mechanism of acetylene hydrogenation. The reactivity and selectivity were used to find the excellent catalysts to promote the hydrogenation reaction. In conclusion, this study broadens the research direction of nickel-based catalysts and also provides some insights into the research in the field of non-precious metals.

## 2. Calculation Methods

The DFT calculation process employed in this investigation was completed in the Guassian09 software package [36]. Based on density functional theory [37], the modeling’s initial configuration was optimized using the B3LYP/6-31G** calculation level. The nonlocal correlation functional and 6-31G** basis set of Lee et al. [38] were applied to C, N, and H atoms, and the Los Alamos effective nuclear pseudopotential (ECP) basis set LANL2DZ was employed for metallic Ni atoms. All optimized structures in the reaction path take into account the Grimme empirical dispersion correction, and the optimized structures’ relative energies are calculated at equal levels from zero-point energy correction [39]. There are no symmetry restrictions during the geometric optimization of the reaction process. The frequency calculation ensures only one hypothetical frequency per transition state and can also be used to check whether the various stabilization points are all minima. The use of Intrinsic Reaction Coordinates (IRC) [40] ensures that the transition state ligation products and reactants are correct as calculations are conducted. Furthermore, the adsorption energies of C_2_H_2_ and H_2_ on graphene-NiN_x_ catalysts were calculated, and the adsorption energies (E_ads_) and co-adsorption energies (E_co-ads_) were defined as in the following:(1)$$E_{\mathrm{ads}}{\;=\;E}_{\mathrm{ads}-\mathrm{state}}-(E_{C_{2}H_{2}{/H}_{2}}{\;+\;E}_{\mathrm{NiNx}})$$
(2)$$E_{\mathrm{Co}-\mathrm{ads}}{=E}_{\mathrm{Co}-\mathrm{ads}-\mathrm{state}}-(E_{C_{2}H_{2}}{+E}_{H_{2}}{+E}_{\mathrm{NiNx}})$$

E_ads_ is single adsorption energy of the reactant molecules on the catalyst. E_co-ads_ are co-adsorption energies. E_ads-state_ is the whole energy of the C_2_H_2_/H_2_ and NiN_x_, E_co-ads-state_ is the overall energy of the system. While $E_{\mathrm{NiNx}}$, $E_{C_{2}H_{2}}$ and $E_{H_{2}}$ are the energies of isolated C_2_H_2_ and H_2_ molecules and NiN_x_, respectively.

## 3. Results and Discussion

### 3.1. Geometries of Reactants

In this study, the optimal graphene-NiN_x_ catalyst and reactants C_2_H_2_ and H_2_ molecules are shown in Figure 1. The optimal graphene-NiN_x_ configurations all have a planar structure, while the C_2_H_2_ molecule is a linear structure. According to the various doping amounts of N atoms, graphene-NiN_x_ can be classified into four geometric configurations: graphene-NiN_1_, graphene-NiN_2_, graphene-NiN_3_, and graphene-NiN_4_, each of which has only one stable configuration. However, Figure 1c–e show that there are three stable geometric configurations of graphene-NiN_2_ (A), graphene-NiN_2_ (B), and graphene-NiN_2_ (C) according to the doping N atoms’ arrangement. Figure 1c–e show that in graphene-NiN_2_ (A) structure, two doped N atoms are in the ortho position in the same five-membered ring; in the graphene-NiN_2_ (B) structure, there are also two doped N atoms. The atoms are located in different five-membered rings on opposite sides of the Ni atom; the two doped N atoms in the graphene-NiN_2_ (C) structure are in the ortho position in the same six-membered ring. For this purpose, the average Ni-N and Ni-C bond lengths of graphene-NiN_x_ (x = 1, 2, 3, 4) were further calculated in this study, and the calculation results are shown in Table 1. In the graphene-NiN_x_ catalyst, the average Ni-N bond length for doping four N atoms is 1.961 Å, which is the maximum in the catalyst. In the graphene-NiN_1_ catalyst, Ni-C is 1.880 Å, which is the maximum data for the average bond length of Ni-C.

Figure 2 shows the electrostatic potential of graphene-NiN_x_ (x = 1, 2, 3, 4) catalysts described on the molecular van der Waals surface (electron density 0.001). The enrichment of positive and negative charges causes a color shift toward blue-red. The higher the shift to blue, the higher the charge density of positive charges, and the higher the shift to red, the higher the charge density of negative charges. As shown in Figure 2, the positive charges of the six catalysts are mainly concentrated near the edges of graphene, as well as on a small number of C atoms distributed at different locations of the catalysts. The Ni atoms and the individual N atoms to which the Ni atoms are bonded have negative charges concentrated on them. It is speculated that the Ni atom may be the reaction’s active site, and it is also proved that the number of added metal Ni atoms and doped N atoms affects the catalyst’s electron distribution.

To deeply understand the catalysts’ properties, the total density of states and the partial density of states of the six catalysts of graphene-NiN_x_ (x = 1, 2, 3, 4) were plotted [41] and added to the corresponding front track, as shown in Figure 3. In the graphene-NiN_1_, the p orbital of C atom and N atom make the main contribution and the secondary contribution, respectively; in the HOMO orbital, the LUMO orbital makes the main contribution is the p orbital of C atom, while the Ni atom makes a greater contribution in an empty orbital. The HOMO orbital contributions of graphene-NiN_3_ and graphene-NiN_4_ have similarities with those of graphene-NiN_1_, and it is evident that the Ni atoms in the LUMO orbitals contribute less to the frontline orbitals. For graphene-NiN_2_ catalysts doped with two N atoms, the p orbital of the C atom in graphene-NiN_2_ (A) makes the primary contributions to both LUMO and HOMO orbital, the p orbital of the N atom is second, and the Ni atom has more contributions. In graphene-NiN_2_ (B) and graphene-NiN_2_ (C), the p orbital of C makes the main contribution to the HOMO orbital, followed by the d orbital of the Ni atom, and finally the p orbital of the N atom. Among the LUMO orbitals, the p orbital of the C atom is the main contributor, and the p orbital of the N atom is the secondary contributor. We speculate that Ni atoms and the C and N atoms coupled to Ni atoms are the catalyst’s potential active sites.

### 3.2. Adsorption of Reactants

In the whole reaction, adsorption is the most crucial step. We place the reactants H_2_ and C_2_H_2_ molecules on the top of the Ni atom, and on the C and N atoms around the Ni atom, to conduct single adsorption behavior, and we obtain the single adsorption structure with the lowest energy. Figure 4 shows the most stable single adsorption structure of the NiN_x_ (x = 1, 2, 3, 4) catalyst. The figure shows that the NiN_x_ series catalysts tend to be adsorbed on the Ni atoms and the C and N atoms are coupled around the Ni atoms. The Ni atom is slightly tilted upwards when the graphene-NiN_4_ catalyst is adsorbing acetylene. Table 2 shows optimal single adsorption and co-adsorption energy of NiN_x_ (x = 1, 2, 3, 4) catalysts through further calculation. The C_2_H_2_ of the six catalysts are all larger than the H_2_ adsorption energy, which proves that during the adsorption, C_2_H_2_ is more likely to be preferentially adsorbed, and then H_2_ is adsorbed to form a co-adsorption structure. Furthermore, the single adsorption energies of both C_2_H_2_ and H_2_ are smaller than the co-adsorption energies ensuring that the reaction is performed within a reasonable range.

### 3.3. Reaction Mechanism of Acetylene Hydrogenation Catalyzed Using Graphene-NiN_x_ Catalysts (x = 1, 2, 3, 4)

In order to fully explore the reaction process of NiN_x_ (x = 1, 2, 3, 4) series catalyst for acetylene hydrogenation, this paper draws a detailed reaction path and the corresponding energy diagram. Graphene-NiN_1_, graphene-NiN_2_ (A), graphene-NiN_2_ (B), graphene-NiN_2_ (C), graphene-NiN_3_, and graphene-NiN_4_ catalysts were investigated for activity and selectivity, respectively.

#### 3.3.1. Reaction Mechanism Utilizing Graphene-NiN_1_

Figure 5 shows configurational changes in each step in the continuous hydrogenation of graphene-NiN_1_ acetylene from acetylene to ethane, while Figure 6 shows the energy corresponding to each step of the reaction pathway. In the optimal co-adsorption of graphene-NiN_1_, the reactants C_2_H_2_ and H_2_ molecules were adsorbed roughly above the catalyst, almost parallel to the graphene-NiN_1_ substrate. First, the reaction pathway starts with co-adsorption (R) with co-adsorption energy of −8.91 kcal/mol. To obtain intermediate IM1, co-adsorption undergoes transition state TS1 with an imaginary frequency of −1831.70 cm^−1^. In TS1, the C_2_H_2_ molecule was deformed, the two H atoms gradually migrated above the Ni atom, and the H1 atom vibrated to the C1 atom to obtain the intermediate IM1. The reaction energy barrier of this process is 28.68 kcal/mol, which is the rate-controlling step for the hydrogenation of acetylene to ethylene. Secondly, the H2 atom coupled to the C3 atom must undergo two transition states TS2 (imaginary frequency of −258.53 cm^−1^) and TS3 (−859.12 cm^−1^) to obtain the intermediate IM3 to produce ethylene. The H2 atom gradually vibrated to the C2 atom during the process from IM2 to TS3 and then to IM3, shortening the bond length from 2.297 Å to 1.082 Å and forming a bond. The energy barriers required for this process are 0.99 kcal/mol and 11.03 kcal/mol, respectively. The co-adsorption structure IM4 was optimized based on IM3 to realize the secondary hydrogenation of acetylene, with the co-adsorption energy of −79.43 kcal/mol. In the co-adsorption of ethylene and hydrogen, ethylene and H3 and H4 molecules are coplanar. Next, IM4 passes through the transition state TS4 with an imaginary frequency of −1652.41 cm^−1^ to obtain the intermediate IM5. The reaction energy barrier for the hydrogenation of ethylene to ethane is 28.38 kcal/mol, and this process is a rate-controlling step. In TS4, two H atoms and ethylene are placed in an oblique line in space. The H3 and H4 atoms migrate slowly to the top of the Ni atom, and the H3 atom stretches toward the C1 atom. The bond length between H3 and C1 is shortened from 1.557 Å to 1.094 Å between TS4 to IM5, and they are connected to form a bond. Finally, the H4 atom attached to C4 approaches the C2 atom and yields the product ethane (P) after passing through the transition state TS5 with an imaginary frequency of −1164.78 cm^−1^. The reaction energy barrier for this step is 6.67 kcal/mol. The bond length of H4 and C2 was shortened from 2.329 Å to 1.094 Å from IM5 to the product ethane and then formed a bond. IRC correctly confirmed the structure before and after the reaction path. It can be concluded from Figure 5 that Ni atoms and surrounding C3 and C4 atoms jointly participate in the catalytic sites of graphene-NiN_1_ catalysts. Figure 6 shows the energy barriers to ethylene and ethane, which are 28.68 kcal/mol and 28.38 kcal/mol, respectively. The two energy barriers are basically the same, so graphene-NiN_1_ has poor selectivity.

#### 3.3.2. Reaction Mechanism Utilizing Graphene-NiN_2_ (A)

Figure 7 and Figure 8 showed the reaction paths and energy diagrams for the continuous hydrogenation of graphene-NiN_2_ (A) acetylene. First, the reaction pathway starts with co-adsorption (R) with co-adsorption energy of −6.26 kcal/mol. It is clear from the co-adsorption graph that the reactants C_2_H_2_ and H_2_ are in the same plane. The co-adsorption to the intermediate IM1 requires the transition state TS1 with an imaginary frequency of −1764.41 cm^−1^, and the reaction energy barrier of this process is 30.14 kcal/mol. This step is a rate-controlling step for the hydrogenation of acetylene to ethylene. The vibration analysis of TS1 also reveals that C_2_H_2_ is deformed, breaking the straight line. The H1 atom vibrates with the C1 atom. In IM1, H1 is bonded to C1, and H2 is bonded to C3 with bond lengths of 1.089 Å and 1.122 Å, respectively. Secondly, due to the long distance between H2 and C2 atoms, IM1 must pass through two transition states to obtain the intermediate ethylene. In TS2 and TS3, only one imaginary frequency is −120.53 cm^−1^ and −729.51 cm^−1^. IM1 only needs to pass through TS2 across an energy barrier of 0.92 kcal/mol to produce the intermediate IM2 that requires one inversion of the vinyl group. Crossing TS3 from IM2 to ethylene requires at least 6.24 kcal/mol of energy. Further ethylene hydrogenation from IM3 yields the co-adsorbed structure IM4, and co-adsorption energy was −64.96 kcal/mol. Next, the reaction undergoes a transition state TS4 with an imaginary frequency of −549.92 cm^−1^ to obtain the intermediate IM5 with a reaction energy barrier of 35.36 kcal/mol. The process of IM5→IM7 requires going through two transition states (TS5 and TS6) with imaginary frequencies of −678.69 cm^−1^ and −1560.44 cm^−1^, respectively. The process is separated into two steps of vibration of the H4 atom: the first step is to attack H4 from C3 to C4, and the second is the H4 atom’s vibration from C4 to N1 atom. The energy barriers required for the two steps are 18.57 kcal/mol and 23.42 kcal/mol. Finally, the reaction requires crossing the energy barrier of 55.38 kcal/mol to generate the product ethane (P) and obtain the transition state TS5 with an imaginary frequency of −1378.67 cm^−1^. This process is a rate-controlling step of the ethylene to ethane hydrogenation process. In TS7, the H4 atom vibrates toward the C2 atom and the bond length is rapidly shortened from 2.251 Å to 1.095 Å. The presence of corresponding reactants and intermediates at each transition state can be confirmed using IRC. Figure 7 concluded that Ni atoms and some C and N atoms around Ni atoms participate in the catalytic active sites of graphene-NiN_2_ (A) catalysts. Figure 8 shows that the energy barrier to forming ethylene is smaller than that of ethane (30.14 kcal/mol < 55.38 kcal/mol). It can be further concluded that graphene-NiN_2_ (A) has certain activity and ethylene selectivity, and is an effective catalyst for acetylene hydrogenation.

#### 3.3.3. Reaction Mechanism Utilizing Graphene-NiN_2_ (B)

Figure 9 and Figure 10 clearly demonstrate the reaction paths and energy diagrams of graphene-NiN_2_ (B) catalysts in the catalytic acetylene hydrogenation process. First, the reaction pathway also starts from co-adsorption (R) with a co-adsorption energy of −6.01 kcal/mol. The process from R to IM1 requires at least 59.05 kcal/mol of energy and experiences TS1 with an imaginary frequency of −1686.31 cm^−1^. This step is a rate-controlled step for the hydrogenation of acetylene to ethylene. In TS1, the H_2_ molecule moves above the N atom, the H1 atom vibrates to the C1 atom. Since the H2 and C2 atoms are far apart, they go through five transition states to form ethylene. IM1 must undergo TS2 with an imaginary frequency of −1522.57 cm^−1^ and overcome the 30.1 kcal/mol energy barrier to produce intermediate IM2. Then, it crosses the energy barrier of 46.4 kcal/mol to reach the intermediate IM3, and there is a transition state TS3 (imaginary frequency is −300.22 cm^−1^) in this process. IM3 undergoes TS4 with an imaginary frequency of −1195.10 cm^−1^ and overcomes the energy barrier of 38.76 kcal/mol to produce intermediate IM4. To generate ethylene, intermediate IM4 also needs to experience TS5 and TS6 with imaginary frequencies of −546.91 cm^−1^ and −893.52 cm^−1^, respectively. The energy barriers required for these two processes are 6.16 kcal/mol and 44.11 kcal/mol, respectively. The H_2_ molecule was placed at a position near ethylene in the IM6 conformation, yielding the co-adsorption structure IM7 with a co-adsorption energy of −52.38 kcal/mol. Subsequently, The H3 and H4 atoms were activated and the bond length expanded to 1.516 Å at one point. The intermediate IM8 was formed through the transition state TS7 with an imaginary frequency of −1791.11 cm^−1^. This process is a rate-controlling step for the hydrogenation of ethylene to ethane, and the reaction energy barrier is 59.80 kcal/mol. The H4 atom connected to the N2 atom needs to undergo TS8 (imaginary frequency of −1529.63 cm^−1^) to obtain the intermediate IM9, and this process must overcome the energy of 13.23 kcal/mol. Next, IM9 needs to undergo two transition states of TS9 and TS10 to form the intermediate IM11, and there is only one imaginary frequency of −408.58 cm^−1^ and −1189.48 cm^−1^, respectively, and the corresponding energy barriers are 43.89 kcal/mol and 42.41 kcal/mol. Finally, the H4 atom attached to N1 only needs to pass through TS11 with an imaginary frequency of −2348.45 cm^−1^ and cross the energy barrier of 11.23 kcal/mol to generate the product ethane. In TS11, the bond length between H4 and C2 is abruptly reduced from 2.337 Å to 1.116 Å. After IRC calculation, it can be confirmed that there are corresponding reactants and intermediates in the transition states of the reaction. It can be concluded from Figure 9 that both Ni atoms and surrounding C and N atoms participate in the catalytic sites of graphene-NiN_2_ (B) catalysts. It can be seen from Figure 10 that the energy barrier for generating ethylene is 59.05 kcal/mol, and the energy barrier for generating ethane is 59.80 kcal/mol, and the difference between the two energy barriers is small. Therefore, the selectivity of graphene-NiN_2_ (B) is not satisfactory.

#### 3.3.4. Reaction Mechanism Utilizing Graphene-NiN_2_ (C)

Figure 11 and Figure 12 show the reaction pathways and energy diagrams for the continuous hydrogenation of graphene-NiN_2_ (C) acetylene. First, the reaction pathway starts with co-adsorption (R) and co-adsorption energy of −6.22 kcal/mol. The intermediate IM1 needs to cross over TS1 with an imaginary frequency of −1678.32 cm^−1^ to obtain the intermediate. The reaction energy barrier of this process is 57.16 kcal/mol. This step is a rate-controlled step for the hydrogenation of acetylene to ethylene. Differently from other catalysts, graphene-NiN_2_ (C) can form ethylene in only one step. In IM1, H1 forms a bond with C1, and H2 forms a bond with C2. The bond lengths are 1.087 Å and 1.086 Å, respectively. IM1 was further hydrogenated to form the most stable and optimized co-adsorption structure IM2, with co-adsorption energy of −51.89 kcal/mol. Secondly, IM2 overcomes the energy barrier of 59.30 kcal/mol and passes through TS2 to obtain the intermediate IM3, which is a rate-controlling step for the hydrogenation of ethylene to ethane. the H3 and H4 atoms continue to vibrate through the N1 atom to C1 in TS2, with an imaginary frequency of −1734.45 cm^−1^. Thirdly, IM3 undergoes a transition state TS3 with an imaginary frequency of −1598.17 cm^−1^ to form the intermediate IM4 with a reaction energy barrier of 15.66 kcal/mol. In TS3, the vibrational direction of the H4 atom is from N1 to N_2_ atom. Finally, IM4 undergoes transition state TS4 with an imaginary frequency of −2386.03 cm^−1^ to form the product ethane with a reaction energy barrier of 56.58 kcal/mol. IRC can confirm that the corresponding reactants and intermediates exist in the reaction’s transition states. Figure 11 concluded that the N1 and N_2_ atoms are involved in the site of the graphene-NiN_2_(C) catalytic reaction. Figure 12 shows that the energy barriers to produce ethylene and ethane are 57.16 kcal/mol and 59.30 kcal/mol, respectively. It can be shown that the two energy barriers are close, and graphene-NiN_2_ (C) has unsatisfactory selectivity.

#### 3.3.5. Reaction Mechanism Utilizing Graphene-NiN_3_

The specific reaction path of graphene-NiN_3_ acetylene continuous hydrogenation and the corresponding energy barrier of each step is as shown in Figure 13 and Figure 14. First, the reaction pathway starts with co-adsorption (R), then the reactants C_2_H_2_ and H_2_ molecules are sterically parallel to the catalyst, and the co-adsorption energy is −54.83 kcal/mol. The transition state TS1 with an imaginary frequency of −1710.68 cm^−1^ is required from the co-adsorption to the intermediate IM1. In TS1, the linear C_2_H_2_ molecule is broken and changed. The H1 and H2 atoms move to the C_2_H_2_ molecule, and the H1 atom undergoes stretching vibration toward the C1 atom to form the intermediate IM1. In IM1, the reaction energy barrier of the process is 39.85 kcal/mol, which is the rate-controlling step for the hydrogenation of acetylene to ethylene, and the bond length of H1 and C1 atoms is 1.094 Å. Secondly, the H2 atom connected to the C3 atom needs to cross the two transition states of TS2 (imaginary frequency of −127.27 cm^−1^) and TS3 (−588.63 cm^−1^) to form the intermediate IM3 to form ethylene. The energy barriers required for these two processes are 0.60 kcal/mol and 3.80 kcal/mol, respectively. Further hydrogenation of acetylene starts from IM4 with a total adsorption energy of −100.52 kcal/mol. In IM4, ethylene and hydrogen molecules are directly above the graphene-NiN_3_ catalyst. Next, to form the intermediate IM5, the reaction undergoes a transition state TS4 with an imaginary frequency of −2173.29 cm^−1^, which is a rate-controlled step for the hydrogenation of ethylene to ethane with a reaction energy barrier of 39.04 kcal/mol. In TS4, the H3 and H4 atoms migrate to the vicinity of the C3 atom, and the H3 atom stretches toward the C1 atom. The reaction’s last step requires the transition state TS5 with an imaginary frequency of −263.51 cm^−1^ to form the product ethane (P). From IM5 to P, the reaction energy barrier for this step is 1.09 kcal/mol. IRC again confirmed the structures’ correctness before and after all transition states in the reaction path. It can be concluded from Figure 13 that C3 atoms are involved in the catalytically active sites of graphene-NiN_3_ catalysts. As can be seen from Figure 14, the energy barrier for the production of ethylene is 39.85 kcal/mol and the energy barrier for the formation of ethane is 39.04 kcal/mol. The reaction energy barriers of the two are basically close, and we can think that the selectivity of graphene-NiN_3_ is poor.

#### 3.3.6. Reaction Mechanism Utilizing Graphene-NiN_4_

Figure 15 and Figure 16 demonstrate the configurational changes and corresponding energies of graphene-NiN_4_ catalysts at each step in the catalytic acetylene hydrogenation process. First, the reaction pathway also starts from co-adsorption (R) with co-adsorption energy of −12.56 kcal/mol. The reactants are free C_2_H_2_ and H_2_ molecules, and their optimal adsorption sites are above the catalyst. The energy of at least 25.24 kcal/mol is required in the process of Co to IM1, which is a rate-controlling step for the hydrogenation of acetylene to ethylene. In the vibrational analysis of TS1, the H_2_ molecule moved above the Ni atom and below the acetylene molecule, and the H1 atom vibrated to the C1 atom, causing the shape of the C_2_H_2_ molecule to change, and only one imaginary frequency value was −1798.29 cm^−1^. Secondly, the H2 atom connected to the metal Ni atom needs to undergo TS2 (imaginary frequency of −1026.95 cm^−1^) to form the intermediate IM2 and must overcome the energy of 20.56 kcal/mol. In TS2, the H2 atom attacks the N1 atom, and the vinyl group undergoes a certain inversion. By the time of the intermediate IM2, the N1 atom forms a bond with the H2 atom with a bond length of 1.035 Å. Then, the H2 atom vibrates toward the C2 atom, the bond length between the H2 and C2 atoms is shortened sharply, and there is only an imaginary frequency with a value of −100.93 cm^−1^ that only needs to cross the energy barrier of 1.25 kcal/mol to form ethylene. Further hydrogenation starts from the co-adsorption structure IM4 with a co-adsorption energy of −59.90 kcal/mol. Subsequently, the H3 and H4 atoms were activated, the bond length was once expanded from 0.742 Å to 1.407 Å, and the intermediate IM5 was formed by the transition state TS4 with an imaginary frequency of −1798.29 cm^−1^. This process is a rate-controlling step for the hydrogenation of ethylene to ethane, and the reaction energy barrier is 51.59 kcal/mol. The last step is the process from IM5 to the product ethane (P), which requires at least 1.43 kcal/mol of energy and crosses the transition state TS5. The H4 atom connected to the N1 atom vibrates toward the C2 atom in the TS5 vibrational analysis, resulting in an associated imaginary frequency (−405.40 cm^−1^). The correct front and rear structures are connected to the transition states in the reaction after IRC calculations. In Figure 15, it can be concluded that Ni atoms and surrounding N1 atoms jointly participate in the catalytic sites of graphene-NiN_4_ catalysts. Figure 16 shows that the energy barrier for the formation of ethylene is smaller than that of ethane (25.24 kcal/mol < 51.59 kcal/mol). We can believe that graphene-NiN_4_ has good reactivity and selectivity, and can be considered a potential catalyst for acetylene hydrogenation.

In the above, the reaction path and energy of the catalyzed by graphene-NiN_1_, graphene-NiN_2_ (A), graphene-NiN_2_ (B), graphene-NiN_2_ (C), graphene-NiN_3,_ and graphene-NiN_4_ catalysts are detailed. To compare the catalytic performance of graphene-NiN_x_ (x = 1, 2, 3, 4) in acetylene hydrogen, the reaction energy barrier of six catalysts are compared, as shown in Table 3. Table 3 shows the order of the energy barriers of graphene-NiN_x_ (x = 1, 2, 3, 4) catalysts to catalyze the hydrogenation of acetylene to ethylene is as follows: graphene-NiN_2_ (B) > graphene-NiN_2_ (C) > graphene-NiN_3_ > graphene-NiN_2_ (A) > graphene-NiN_1_ > graphene-NiN_4_. The order of selectivity in Table 3 is graphene-NiN_4_ > graphene-NiN_2_ (A) > graphene-NiN_2_ (C) > graphene-NiN_3_ > graphene-NiN_2_ (B) > graphene-NiN_1_. The catalytic performance of catalysts depends on both reaction activity and selectivity. The lower the activity of the catalyst and the higher the selectivity, the better the catalytic performance. It is clearly seen that graphene-NiN_4_ has excellent catalytic properties, followed by graphene-NiN_2_ (A).

Thus, we also speculate that the catalyst doped with even N atoms makes the catalyst structure distribution more uniform, enhances the overall symmetry, and is more conducive to the hydrogenation of acetylene, whereas the catalyst doped with odd-numbered N atoms is the opposite, which is unfavorable for the acetylene hydrogenation reaction.

## 4. Conclusions

In this study, the reaction mechanism of acetylene hydrogenation catalyzed by graphene-NiN_x_ (x = 1, 2, 3, 4) catalyst doped with a various number of N atoms was studied using the theoretical approach of DFT. The results reveal that the order of energy barrier for ethylene formation from acetylene hydrogenation catalyzed using graphene-NiN_x_ (x = 1, 2, 3, 4) catalyst is graphene-NiN_2_ (B) > graphene-NiN_2_ (C) > graphene-NiN_3_ > graphene-NiN_2_ (A) > graphene-NiN_1_ > graphene-NiN_4_. Among the six catalysts, graphene-NiN_4_ has the highest selectivity, followed by graphene-NiN_2_ (A), and the rest are less selective. Having low activity and high selectivity is evidence of the good catalytic performance of this catalyst. Thus, graphene-NiN_4_ exhibits excellent catalytic performance and is also a potential non-noble metal catalyst for acetylene hydrogenation.

## Figures and Tables

**Figure 1 molecules-27-05437-f001:**
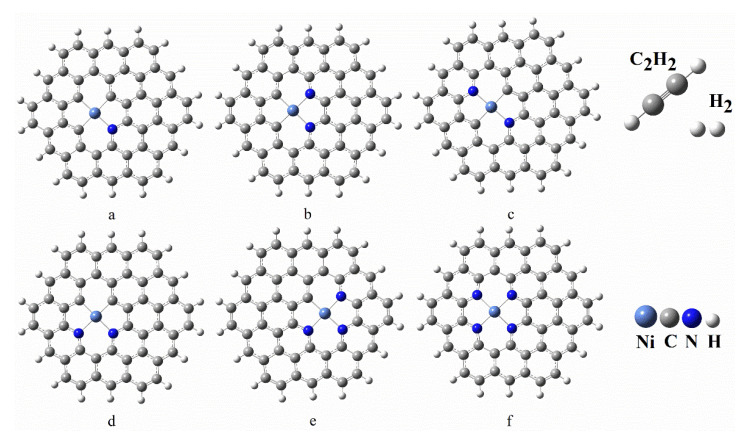
Optimized structures of C_2_H_2_, H_2_, graphene-NiN_1_ (**a**), graphene-NiN_2_(A) (**b**), graphene-NiN_2_(B) (**c**), graphene-NiN_2_(C) (**d**), graphene-NiN_3_ (**e**), and graphene-NiN_4_ (**f**).

**Figure 2 molecules-27-05437-f002:**
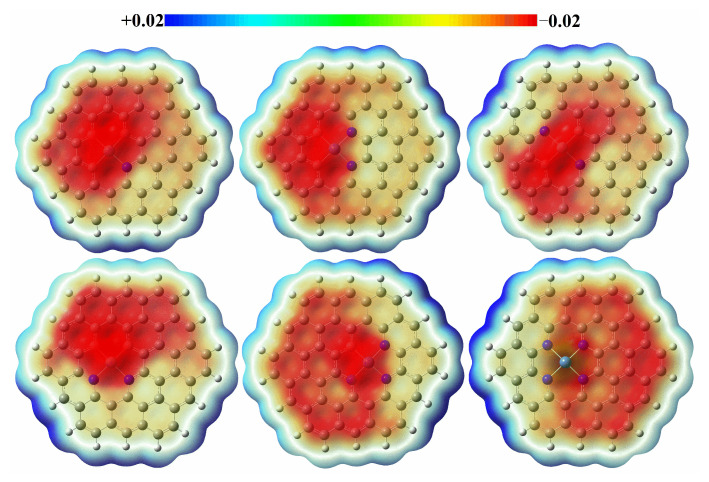
The molecular electrostatic potential of graphene-NiN_1_, graphene-NiN_2_ (A), graphene-NiN_2_ (B), graphene-NiN_2_ (C), graphene-NiN_3_ and graphene-NiN_4_ (electron density: 0.001 a.u.).

**Figure 3 molecules-27-05437-f003:**
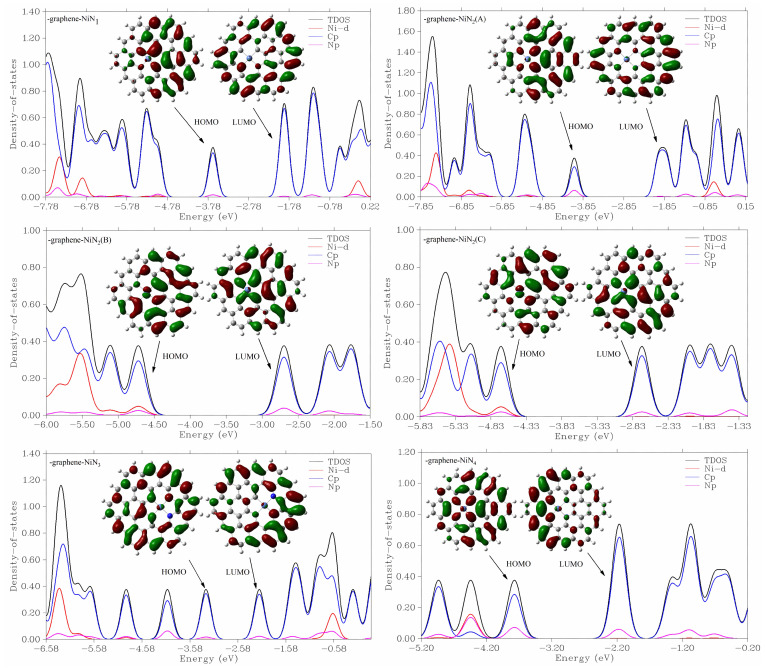
The DOS of graphene-NiN_1_, graphene-NiN_2_ (A), graphene-NiN_2_ (B), graphene-NiN_2_ (C), graphene-NiN_3_ and graphene-NiN_4_, and corresponding HOMO and LUMO.

**Figure 4 molecules-27-05437-f004:**
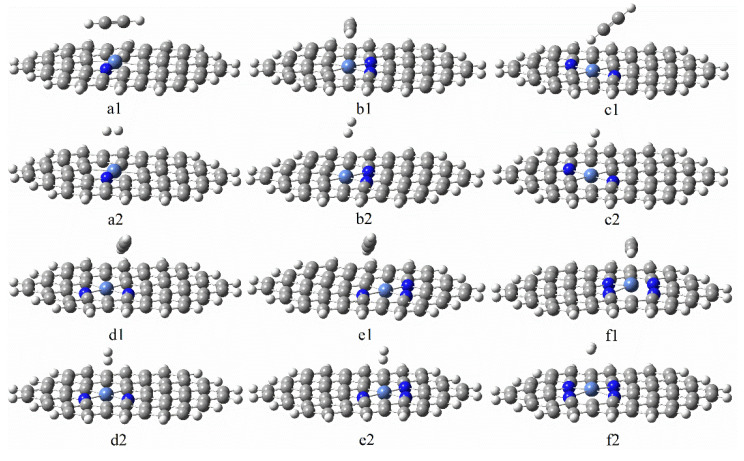
The most stable adsorption structures of C_2_H_2_ and H_2_ on graphene-NiN_1_ (**a1**,**a2**), graphene-NiN_2_ (A) (**b1**,**b2**), graphene-NiN_2_ (B) (**c1**,**c2**), graphene-NiN_2_ (C) (**d1**,**d2**), graphene-NiN_3_ (**e1**,**e2**), and graphene-NiN_4_ (**f1**,**f2**).

**Figure 5 molecules-27-05437-f005:**
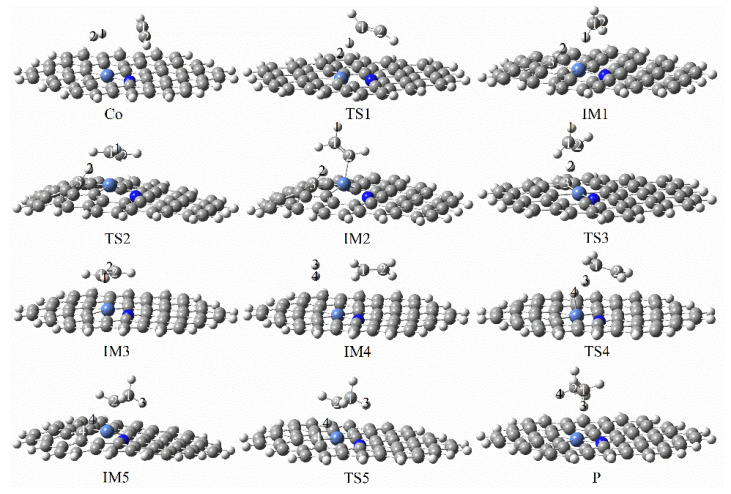
Optimal reaction pathway for acetylene hydrogenation catalyzed by graphene-NiN_1_ catalyst.

**Figure 6 molecules-27-05437-f006:**
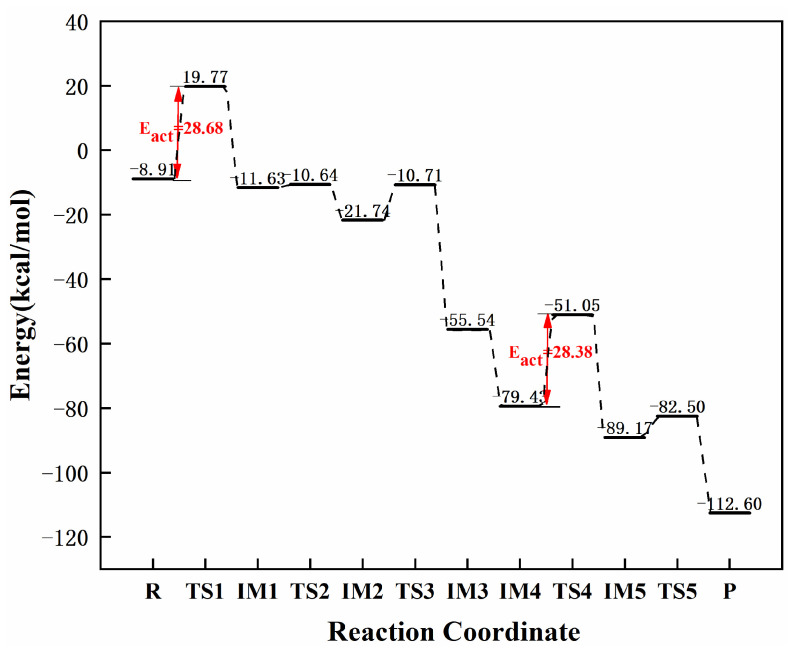
The most favorable energy path diagram for the hydrogenation of acetylene on graphene-NiN_1_.

**Figure 7 molecules-27-05437-f007:**
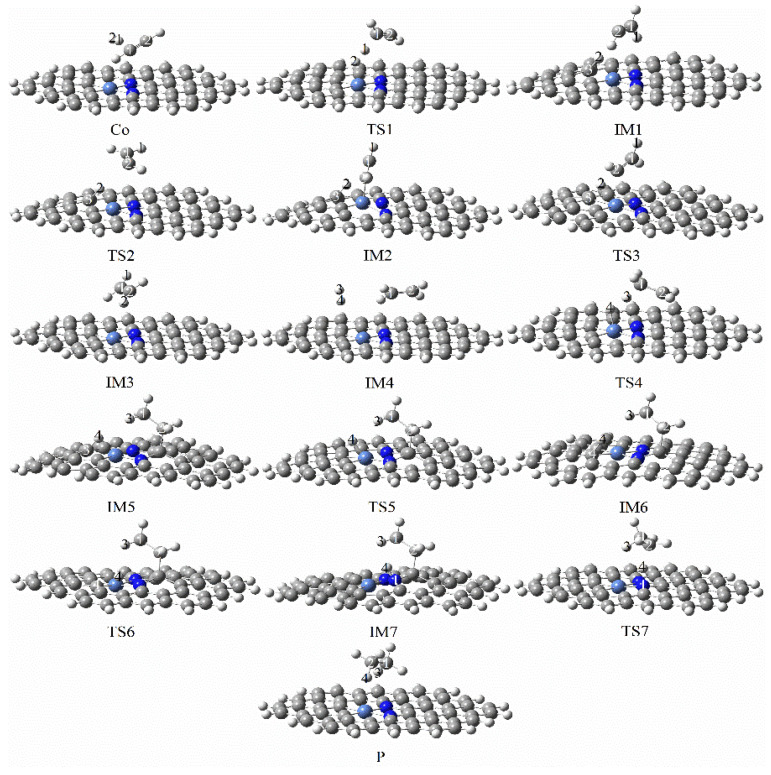
Optimal reaction pathway for acetylene hydrogenation catalyzed by graphene-NiN_2_ (A) catalyst.

**Figure 8 molecules-27-05437-f008:**
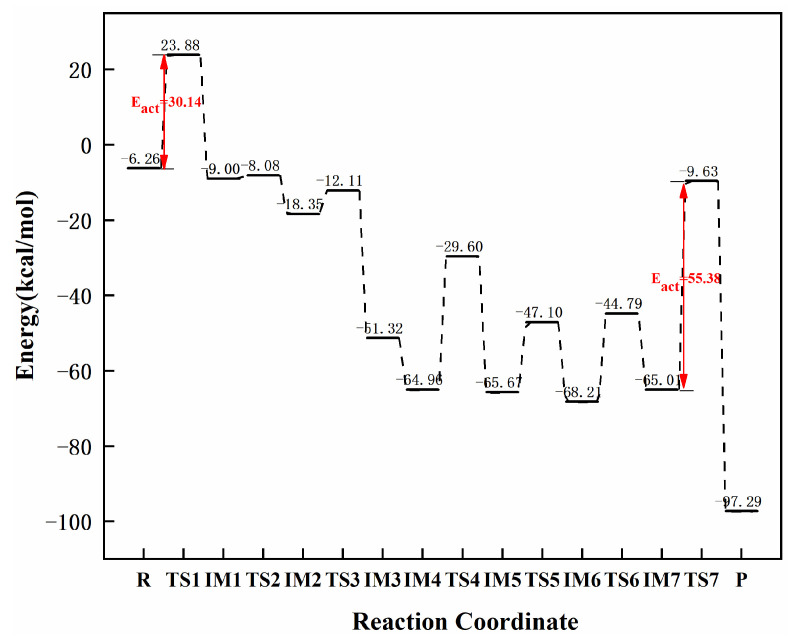
The most favorable energy path diagram for the hydrogenation of acetylene on graphene-NiN_2_ (A).

**Figure 9 molecules-27-05437-f009:**
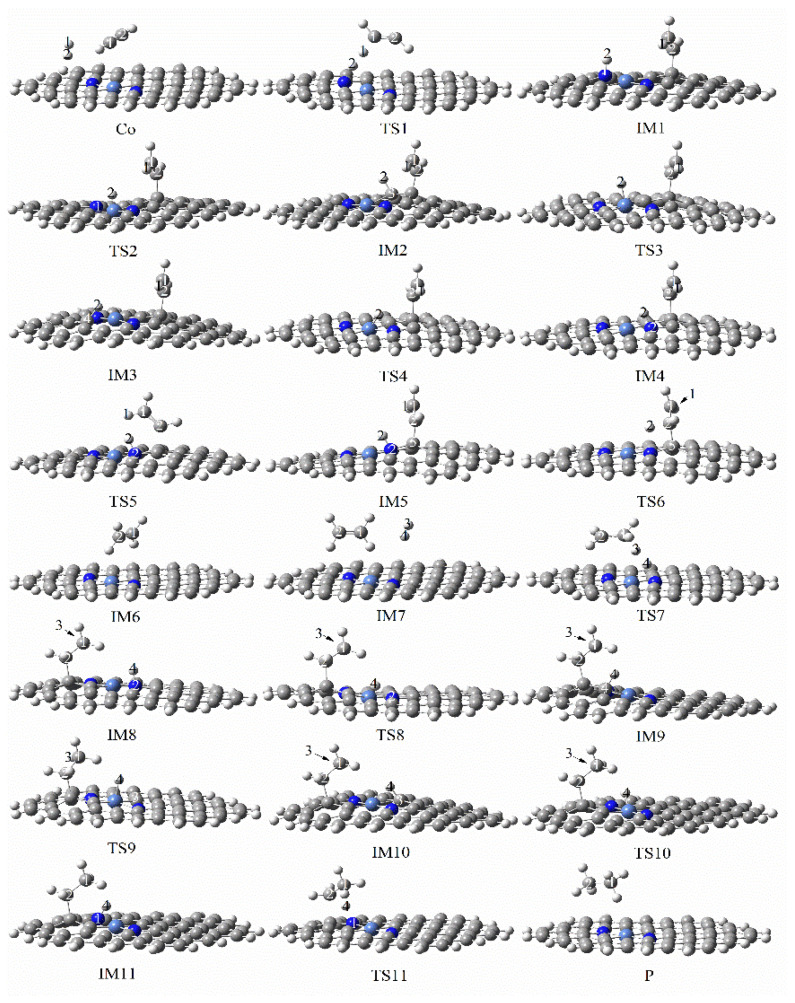
Optimal reaction pathway for acetylene hydrogenation catalyzed by graphene-NiN_2_ (B) catalyst.

**Figure 10 molecules-27-05437-f010:**
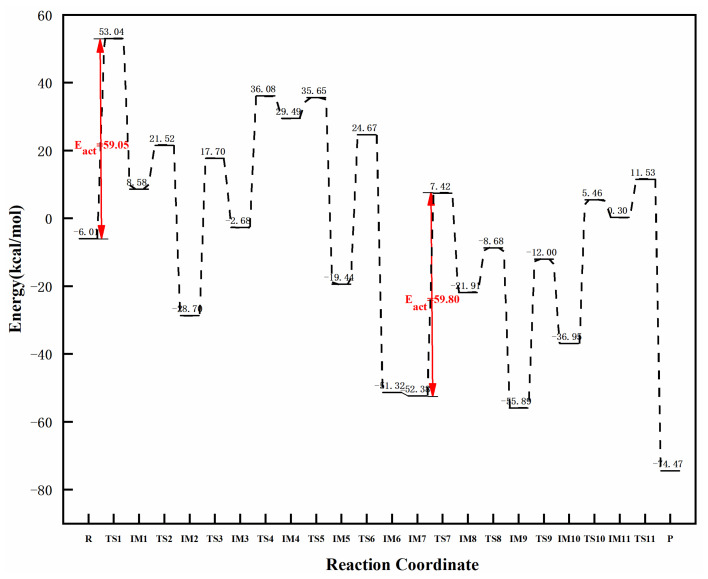
The most favorable energy path diagram for the hydrogenation of acetylene on graphene-NiN_2_(B).

**Figure 11 molecules-27-05437-f011:**
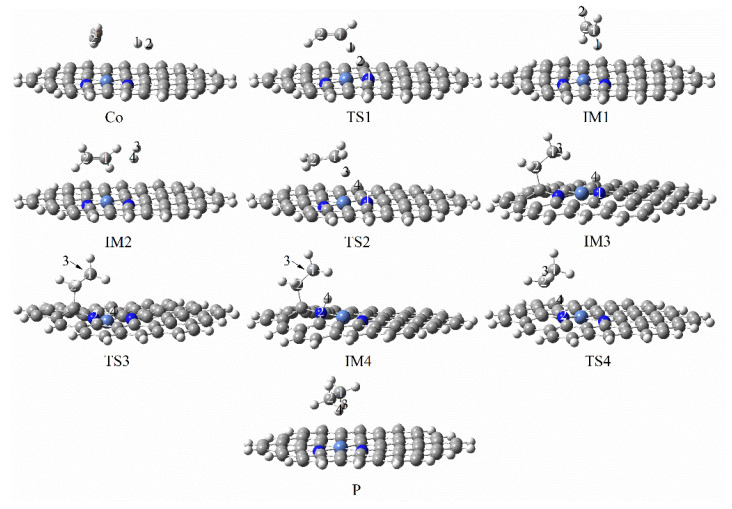
Optimal reaction pathway for acetylene hydrogenation catalyzed by graphene-NiN_2_(C) catalyst.

**Figure 12 molecules-27-05437-f012:**
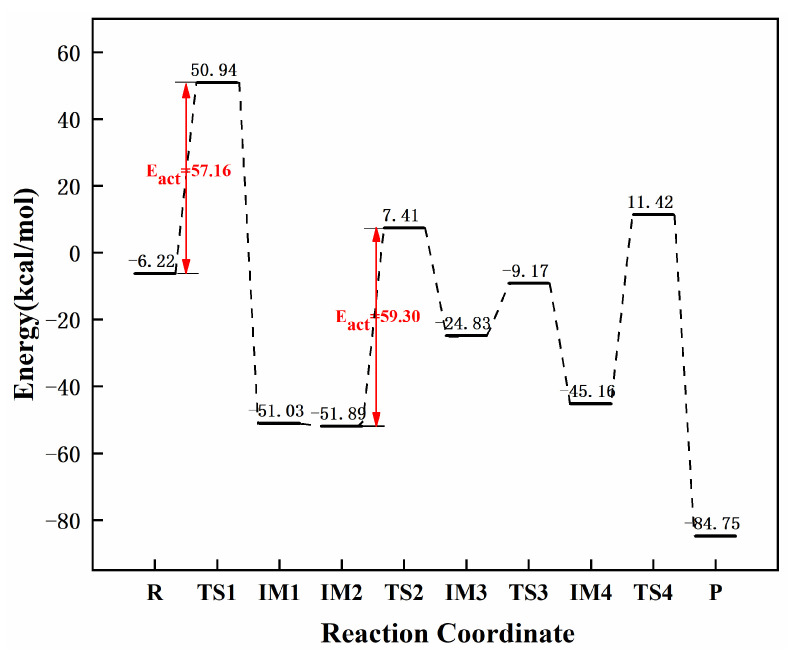
The most favorable energy path diagram for the hydrogenation of acetylene on graphene-NiN_2_(C).

**Figure 13 molecules-27-05437-f013:**
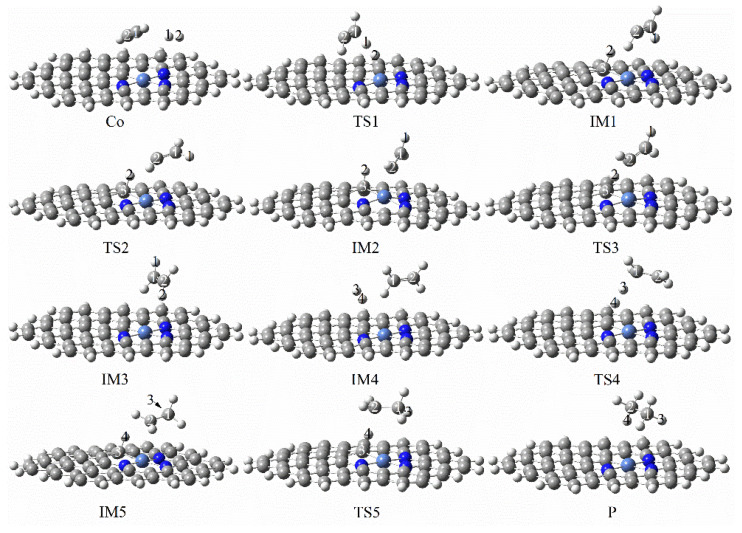
Optimal reaction pathway for acetylene hydrogenation catalyzed by graphene-NiN_3_ catalyst.

**Figure 14 molecules-27-05437-f014:**
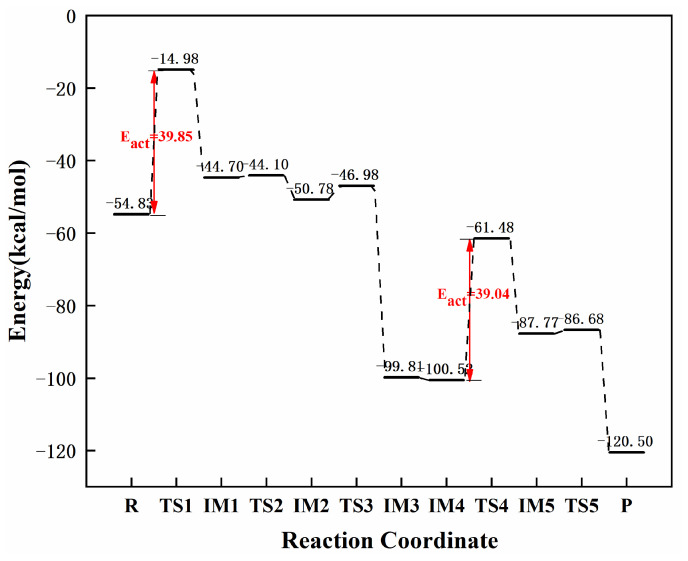
The most favorable energy path diagram for the hydrogenation of acetylene on graphene-NiN_3_.

**Figure 15 molecules-27-05437-f015:**
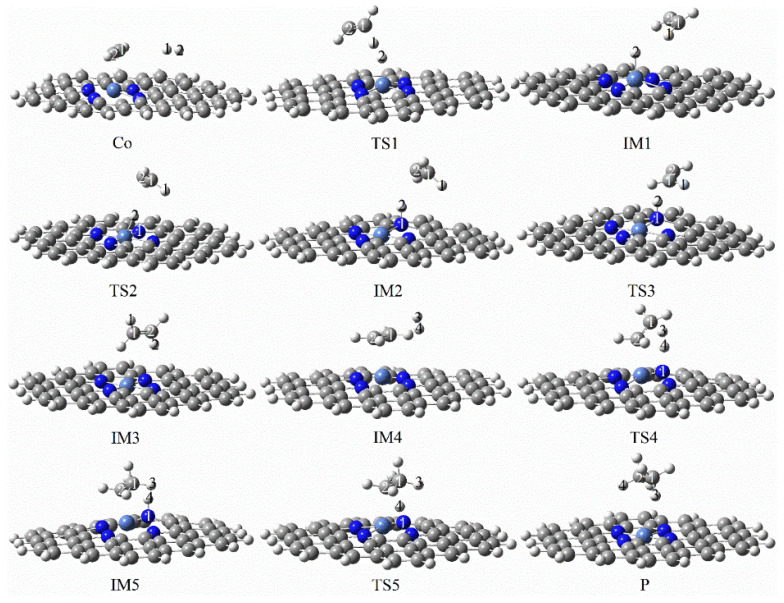
Optimal reaction pathway for acetylene hydrogenation catalyzed by graphene-NiN_4_ catalyst.

**Figure 16 molecules-27-05437-f016:**
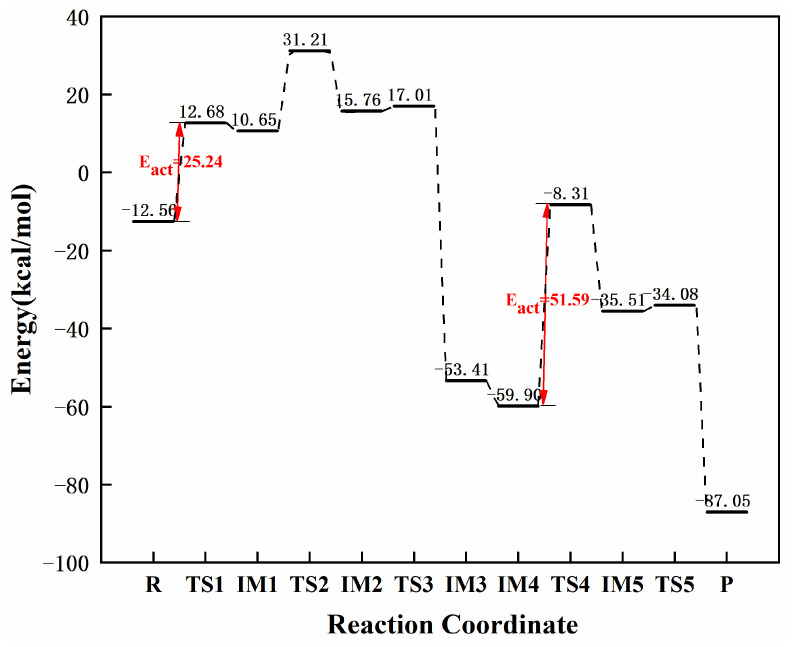
The most favorable energy path diagram for the hydrogenation of acetylene on graphene-NiN_4_.

**Table 1 molecules-27-05437-t001:** Average bond lengths of Ni–N and Ni–C for graphene-NiN_x_ catalysts (x = 1, 2, 3, 4) (unit: Å).

	d (Ni–N)	d (Ni–C)
graphene-NiN_1_	1.951	1.880
graphene-NiN_2_ (A)	1.922	1.871
graphene-NiN_2_ (B)	1.932	1.872
graphene-NiN_2_ (C)	1.951	1.863
graphene-NiN_3_	1.909	1.862
graphene-NiN_4_	1.961	--

**Table 2 molecules-27-05437-t002:** Single and co-adsorption energies of C_2_H_2_ and H_2_ adsorbed on graphene-NiN_x_ (x = 1, 2, 3, 4) catalysts (energy unit: kcal/mol).

Catalyst	C_2_H_2_	H_2_	Co-Adsorption
graphene-NiN_1_	−8.43	−0.60	−8.91
graphene-NiN_2_ (A)	−4.11	−1.23	−6.26
graphene-NiN_2_ (B)	−5.22	−1.35	−6.01
graphene-NiN_2_ (C)	−5.29	−1.42	−6.22
graphene-NiN_3_	−53.83	−50.13	−54.83
graphene-NiN_4_	−11.55	−1.79	−12.56

**Table 3 molecules-27-05437-t003:** Rate-controlling step energy barriers for acetylene hydrogenation catalyzed by graphene-NiN_x_ (x = 1, 2, 3, 4) catalysts (energy unit: kcal/mol).

Catalyst	E_ethylene barrier_	E_ethane barrier_	Selectivity	Source
graphene-NiN_1_	28.68	28.38	0.30	this work
graphene-NiN_2_ (A)	30.14	55.38	25.24	this work
graphene-NiN_2_ (B)	59.05	59.80	0.75	this work
graphene-NiN_2_ (C)	57.16	59.30	2.14	this work
graphene-NiN_3_	39.85	39.04	0.81	this work
graphene-NiN_4_	25.24	51.59	26.35	this work

## Data Availability

All data included in this study are available upon request by contact with the corresponding author.

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
