# Peer review of "Density Functional Theory Study on NiNx (x = 1, 2, 3, 4) Catalytic Hydrogenation of Acetylene"

_molecules, 2022, doi:10.3390/molecules27175437_

Round 1
Reviewer 1 Report
Please see the attachment.

Reviewer 2 Report
Dear editor
Hou and Kang performed a theoretical study on a series of catalytic materials based on doped-Ni graphene for hydrogenation of acetylene. The manuscript is understandable and presents a thorough study based on IRC calculation to show the reactive route of acetylene. Nevertheless, some issues require to be attended prior to publication.
Major revision:
1.The main objective of the work is not defined. Please define it at the end of the introduction.
2. In the computational details, it is not justified why B3LYP was used. A benchmark study is highly recommended.
3. The concept of ‘co-adsorption’ is necessary to be defined, previously to be given in Eq. 2.
4. What is the physical meaning of the computed DOS, if all systems under study are molecules? That is, the DOS are strictly defined for solid state systems.
5. To test the validity of the molecular model systems, the present results should be compared to the DOS obtained with periodic conditions.
6. The paragraph on page 3 reading: “The contribution of the HOMO orbital is consistent, while the contribution of Ni atoms in the empty orbital is more”, is ambiguous. Please revise.
7. In Fig. 3, the quantities of the graph and the legends should be improved. They seem to be screen snapshots.
8. Also in Fig.3, what can be said about the size of the band gaps, and the possible catalyst performance?
9. As a general remark, all energy barriers for the model systems under study, except those given in Fig. 6, are quite large.
The authors are requested to find experimental data to evaluate the feasibility of the reactions, since they appear to be improbable.
10. It recommended to shorten the discussion of each of the reaction steps and models, since it is redundant and quite hard to follow the reading. Only important results should be given.
11. In the conclusion section, the authors should enunciate which models are more susceptible to be used as catalysts in the light of the revision of results, as requested for the size of the energy barriers.
Minor revision
1. Some typos were identified such as ‘minimums’, which should be ‘minima’. Revise overall English style throughout the manuscript.
Reviewer 3 Report
Referee report
Density Functional Theory Study on NiNx(x=1,2,3,4) Catalytic Hydrogenation of Acetylene
Cuili Hou, Lihua Kang
1. In the abstract it is not clear how the physical result “The calculation results reveal that the graphene-NiN4 has better catalytic performance” is connected with numerical results of the paper. However there is a very long enumeration of standard computational technique (This comment is also related to the beginning of section 2)
2. Abbreviations STEM, FTIR, are not explained
3. The authors performed many calculations by reliable program software, enumerate a lot of intermediate results, but the connection between this values and conclusion is not seen.
4. In conclusion The results reveal that the order of energy barrier for ethylene formation from acetylene hydrogenation catalyzed using graphene-NiNx (x = 1, 2, 3, 4) catalyst is graphene-NiN4 < graphene-NiN1 < graphene-NiN2(A) < graphene-NiN3 < graphene-NiN2(C) < graphene-NiN2(B), and the order of energy barrier for ethane formation is graphene-NiN1 < graphene-NiN3 < graphene-NiN4 < graphene-NiN2(A) < gra-phene-NiN2(C) < graphene-NiN2(B). Phrase looks not finished to what is related the word “that ”?
5. However the next sentence claims so many properties, which depend on many parameters and can be verified just experimentally “Among the graphene-NiNx series catalysts, graphene-NiN4 exhibits excellent catalytic performance, low activity, and high selectivity. It is a potential non-noble metal catalyst for acetylene hydrogenation”.
